# The Role of Anharmonicity in (Anti-)Ferroelectric Alkali Niobates

**DOI:** 10.3390/ma18194593

**Published:** 2025-10-03

**Authors:** Leif Carstensen, Wolfgang Donner

**Affiliations:** Department of Materials and Earth Sciences, Technische Universität Darmstadt, 64287 Darmstadt, Germany

**Keywords:** antiferroelectric, anharmonic, niobate, electron density, perovskite

## Abstract

NaNbO_3_ (NN), known for the complexity of its phase transition sequence, is antiferroelectric (AFE) at room temperature, while both LiNbO_3_ (LN) and KNbO_3_ (KN) are ferroelectric (FE). The origin of ferroelectricity in ABO_3_ perovskites is believed to lie in the B-O hybridization, but the origin of antiferroelectricity remains unclear. Recent ab initio studies have shown that the same B-O hybridization is necessary in AFE and proposed an additional, anharmonic contribution to the potential of the A-site atom as the crucial difference between FE and AFE perovskites. We used structure factors obtained from X-ray diffraction experiments in combination with the Maximum Entropy Method to obtain electron densities for LN, KN, and NN and identify differences in their bonding behavior. We present experimental evidence for anharmonic A-site contributions of varying strength in alkali niobates, pointing at a new path for the design of (anti-)ferroelectric materials.

## 1. Introduction

There have been competing theories about the nature of ferroelectricity in ABO_3_ perovskites in the past. The Goldschmidt tolerance factor *t* describes the ratio of ionic radii to determine the crystal structure and is thus often used to predict ferroelectricity, but fails to explain the phase transitions. Bersuker et al. proposed that the Pseudo-Jahn-Teller effect (PJTE) is sufficient to explain the structural changes in a cubic-tetragonal-orthorhombic or cubic-rhombohedral phase transition sequence, whereas Cohen argued that long range coulomb interaction is required on top of the B-O hybridization [1,2]. However, the theories agree that the origin of ferroelectricty in perovskite materials lies in the hybridization of B-site cations and oxygen anions. The majority of perovskite materials exhibit ferroelectric (FE) or paraelectric properties at room temperature (RT), but there are a few materials that show antiferroelectric (AFE) properties, e.g., PbZrO_3_ (PZ) and NaNbO_3_ (NN) [3,4]. PZ has been shown to exhibit A-site disorder and the lone electron pair commonly associated with lead, possibly linking the A-site to the AFE nature of the material [5,6], but NN is not known to exhibit either of these features.

The theory of antiferroelectricity has captivated the attention of researchers for the past two decades, with studies being dedicated to understanding the AFE state [7] or PZ as a prototype AFE system [8], as well as to finding new AFE materials [9,10] or compositions [11,12]. Structurally, the main difference between ferroelectrics and antiferroelectrics is the absence of an inversion center in the former and the presence of such in the latter. In the case of the alkali niobates, this inversion center lies in a plane containing A-O bonds. While the authors of the theories about ferroelectricity mentioned above did not agree on every detail, the hybridization of B and O atoms and its contribution to ferroelectricity in classical ABO_3_ perovskites was agreed upon. The hybridization of occupied O_p_ orbitals with unoccupied B_d_ orbitals occurs in ferroelectric perovskites to decrease electronic state degeneracy and yields a covalent bond character [13]. The Nb-O hybridization changes the potential energy surface, causing a freezing of certain phonons in specific positions, i.e., rotations and distortions of the Nb-O octahedra, to form a FE structure. It is reasonable to assume that the same requirement is valid for antiferroelectricity, thus B-O hybridization could be viewed as a necessary condition for antiferroelectricity [14]. In the case of NN, this assumption has been confirmed by Hadaeghi et al. who, following Bersukers Pseudo-Jahn-Teller effect theory, found coupling between the respective Nb and O orbitals in both FE and AFE NN in ab initio studies. They found the $\Delta_{5}$ phonon mode to be crucial for the antiferroelectricity in NN and showed that the FE and AFE phases of NN exhibit different biquadratic coupling of the phonon modes creating the FE or AFE instability, raising the question of which effect is causing the coupling to change in the energetically favored AFE structure [15]. Hence the specific origin of antiferroelectricity (AFE) in NaNbO_3_ has yet to be found, but the lack of other chemical differences between the alkali niobates LiNbO_3_ (LN) (FE, rhombohedral), NaNbO_3_ (AFE, orthorhombic), and KNbO_3_ (KN) (FE, orthorhombic) inevitably shifts the attention to the A-site. This assumption was confirmed by Yoneda et al., who found differences in the XAFS patterns of NaNbO_3_, KNbO_3_, and AgNbO_3_ related to the A-site cations [16]. The different XAFS patterns are an expression of different potentials between the A and O species in those systems. First principles research on the effect of A-site cations of PZ (AFE, orthorhombic) and CaTiO_3_ (FE, orthorhombic) has shown a tendency of a modified CaTiO_3_ to become antiferroelectric upon adding a specific anharmonic term to the Ca potential [17]. This reveals once again the importance of the A-site in antiferroelectricity and narrows down the focus to anharmonic A-O interactions.

The electron density (ED) distribution inside the unit cell can be expanded into a Fourier series, as shown in Equation (Equation 1), where the Fourier coefficients are directly related to the x-ray structure factors by(1)$$\rho \left(\vec{r}\right)=\sum F\left(\vec{H}\right)exp\left(-2\pi i\vec{H}\vec{r}\right),$$
with the electron density ρ, the coordinate vector inside the unit cell $\vec{r}\left(x,y,z\right)$, the scattering vector of a reflection $\vec{H}\left(h,k,l\right)$ and the structure factor $F(\vec{H})$. However, experimentally only a limited number of structure factors can be determined through the intensity of X-ray diffraction. It was shown by Collins in 1982 that it is possible to obtain ED distributions from imperfect data sets [18]. The electron density was used as a proxy for the interatomic potential in a study of the cubic phases of NN, KN, and their solid solution [19]. The authors used Rietveld refinement to extract structure factors from X-ray powder diffraction experiments and determined the electron density in the unit cells by the Maximum Entropy Method (MEM). The study revealed no significant anharmonic A-O bonding features, but it was restricted to the cubic high-temperature phases, whereas a study of the room temperature (RT) phases is lacking. Calculating the exact electron density distribution requires a complete set of infinitely well resolved structure factors, which is impossible to obtain experimentally. The Maximum Entropy algorithm is designed to gain the most probable result from an incomplete set of data by maximizing an entropy-like value for the probabilities left undefined by the data. Here, the unit cell is divided into voxels with a uniform MEM electron density across the entire cell and the algorithm shifts MEM electron density towards voxels at positions of overlap between strong structure factors [20]. The DYSNOMIA software package performs a fast Fourier transformation of the structure factor data, comparing each new voxel set to the last, as well as to the initial set of structure factors [21]. The results of MEM heavily depend on the quality of the dataset, thus the quality of the Rietveld refinement is crucial to a successful electron density calculation via MEM. A study by Tanaka et al. used the MEM based ED to generate the unobserved structure factors needed for an accurate calculation of the electrostatic potential [22]. The electronic contribution $U_{ele}$ to the electrostatic potential is given in Equation (Equation 2)(2)$$U_{ele}\left(\vec{r}\right)=-4\pi \sum_{\vec{H}}\frac{\tilde{\rho }\left(\vec{H}\right)exp\left(i\vec{H}\vec{r}\right)}{|\vec{H}{|}^{2}},$$
where $\tilde{\rho }$, $\vec{H}$ and $\vec{r}$ denote the Fourier component of the electron density ρ, the scattering vector and the coordinate vector, as in Equation (Equation 1).

This work uses the Rietveld/MEM approach to take a closer look at the bonding features in the ED of the RT phases of LN, KN, and NN, revealing experimental evidence for the anharmonic interaction of A-site atoms with the surrounding O ions and leading to the conclusion that a sufficient condition for antiferroelectricity can be found within the details of the A-O interaction.

## 2. Materials and Methods

We used single crystals of LN (Alineason Materials Technology GmbH, Frankfurt am Main, Germany, Czochralski), KN (FEE GmbH, Idar-Oberstein, Germany, top-seeded solution growth), and P-phase NN (self-grown using a flux method detailed by Dec and Zhelnova [23,24]) and ground them to powders to minimize statistical errors caused by the strong twinning tendencies of NN which could hamper the extraction of structure factors. The peak overlap caused by twinning still exists in powder diffraction, but it is mitigated by a much better statistical average. The powder diffraction experiments were carried out using Mo-K_α1_ radiation on a STOE Stadi P diffractometer in transmission geometry, a Ge(220) monochromator, a rotating glass capillary sample holder and a DECTRIS Mythen 1K line detector. Data were collected up to a momentum transfer of Q ≈ 8.86 Å^−1^ with a step size of 0.2° to mitigate otherwise necessary flat field and detector specific position sensitivity corrections. Measured data (ΔQ ≈ 0.0025 Å^−1^) was summed up with a binning size of 0.02°, corresponding to a final data resolution of ΔQ ≈ 0.003 Å^−1^. The use of a capillary instead of a flat plate sample holder requires the geometric corrections to be applied according to a Debye–Scherrer geometry instead of the standard STOE geometry. The process of structure factor extraction was performed by careful Rietveld refinement using Debye–Waller (DW) factors and atomic positions obtained from neutron or synchrotron diffraction literature data [25,26]. Variations of the Nb position and DW factor had no visible impact on the refinement or the subsequent ED calculation and were thus omitted. For the Rietveld refinement, we used geometrically obtained axial divergence parameters, instrumental resolution functions based on LaB_6_ calibration measurements and absorption corrections with respect to the capillary diameter and sample material with an estimated packing density of 0.6.

Using corrections based on the described geometry and measurement process, a Rietveld refinement calculates the structure factors for all measured reflections, which can be used to calculate the electron density in the unit cell via a Fourier expansion, as shown in Equation (Equation 1). This is possible because the intensity of any reflection $I\propto F^{2}\left(hkl\right)$ is proportional to the square of its structure factor *F(hkl)*, which depends on the atomic positions *(x,y,z)* in the unit cell. As shown in Equations (Equation 3) and (Equation 4), the structure factor of any reflection *(hkl)* is proportional to the atomic form factors *f_j_*, which in turn are proportional to the electron density $\rho (\vec{r^{\prime }})$, where $\vec{r^{\prime }}$ represents a vector originating from the scattering atom and Q represents the momentum transfer of the scattering event.(3)$$F\left(hkl\right)=\sum_{j}f_{j}\left(Q\right)e^{-2\pi i(hx+ky+lz)}$$(4)$$f\left(Q\right)=\int \rho \left(\vec{r^{\prime }}\right)e^{iQ\vec{r^{\prime }}}d^{3}\vec{r^{\prime }}$$

Regarding the control parameters of the MEM calculation (DYSNOMIA 1.0), the structure factor uncertainty, E, was kept between 0.1 and 0.5. Generalized constraint parameter weighting factors $\lambda_{n}$ were adjusted as needed, using $\lambda_{4}=1$ constraints for all materials with $\lambda_{6}=1$ used as needed. The weighting factor x, implemented to decrease the noise caused by strong reflections, was varied between 2, as suggested for powder diffraction, and 3, as 4 is suggested for single crystals [27]. A comparison of x = 2 and x = 4 for NN can be found in Figure A1 in the Appendix A. Visualization of ED maps and line profiles, calculations of ED and .cif-related data were performed using VESTA (Version 3.5.8) [28]. Line profiles were defined peak to peak in the ED.

## 3. Results

Rietveld profiles of LiNbO_3_, KNbO_3_, and NaNbO_3_ are shown in Figure 1, Figure 2 and Figure 3. The intensity axes are set to logarithmic for better visibility of small reflections, difference plots are kept linear to avoid overlap. The Rietveld profiles show that the powders are of high purity, as there are no observable secondary phase reflections. Using no instrumental resolution function (IRF) would improve the reliability factors of the NN fit to R ≈ 6.8 and R_f_≈ 4.2, but the intention behind using an IRF is to keep the fit as physically reasonable as possible, in order to avoid generation of artifacts in the MEM calculations.

The structure factors obtained from the Rietveld refinements shown in Figure 1, Figure 2 and Figure 3 are available in the data repository listed below. An overview of the MEM electron density distributions of LN, KN, and NN is shown in Figure 4. All three materials show well defined bonds in the Nb-O octahedron and a mostly spherical A-site atom at an isosurface level of 1 e^−^/Å^3^. Voronoi integrations of the unit cells yield between 35 e^−^ and 38 e^−^ for the Nb ions, hinting at the covalent character of their bonds. Figure 5 and Figure 6 represent atom to atom line profiles derived from Figure 4, showing the numerical values of MEM electron density along the bonds, resulting from a voxel size of about 10^−5^ Å^3^. A closer look at the Nb-O bonds in Figure 5 reveals that the bond strength minimum tends to be slightly closer to the O position than the Nb position. The bond strength minima are quite shallow at about 1 MEM ED unit, again mirroring the covalency of the hybridized Nb-O bonds. A harmonic covalent bond is expected to resemble a parabola, which is slightly distorted due to the different core electron densities of the respective ions. An exemplary harmonic fit for the long Nb-O bond in LN is shown in green in Figure 5a. Keep in mind that the harmonic approximation only works for small displacements, thus the fit is expected to deviate from the bond profile. Deviations from the harmonic shape are marked with an asterisk and will be analyzed in more detail in the next section. The KN Nb-O bonds are the most harmonic bonds found in this study, whereas the short Nb-O bond in LN exhibits a small shoulder on the Nb side of the bond. All Nb-O bonds in NN feature a similar anomaly, which is most pronounced in the short Nb-O4 bond, as well as the Nb-O1 and Nb-O2 bonds. The shapes of the core MEM electron density distributions appear symmetric for all three materials, as can be seen by comparing the slopes for the different length bonds of each atom, representing O (1, 2, and 4 different positions for LN, KN, and NN, respectively) on the left and Nb on the right side.

The A-O bonds are much more variable in bond length and are expected to be more ionic (deeper) than the Nb-O bonds. The PJTE induced breaking of symmetry causes a variety of bond lengths, which will be described in this paragraph. Starting with LN in Figure 6a, there are four distinct bonds between Li and O, with bond lengths of approximately 2 Å for short bonds and 3.3 Å for long bonds. Deviations from a harmonic bond shape are only visible below 1 MEM ED unit. Three Li-O bonds exhibit a shoulder on the O side, while the second longest bond exhibits a shoulder on the Li side. Additionally, the longest bond exhibits a local maximum approximately 1.7 Å from the oxygen position. KN has three distinct K-O1 bonds and two distinct K-O2 bonds, which can be seen in Figure 6b. Similarly to LN, the bond with the lowest minimum value is not the longest, but the second longest K-O1 bond. This bond also features a shoulder on the K side, which is not found in the other two K-O1 bonds. The shoulder is also found in both K-O2 bonds in Figure 6b. Figure 6c,d show the different Na-O bonds in NN, which features a total of 15 distinct bonds, seven for Na1-O*x* and eight for Na2-O*x*. The Na1-O1 bonds in Figure 6c show an unremarkable curve for the short bond, but the intermediate bond exhibits a linear increase towards the O1 position, whereas the long bond exhibits a shoulder towards the Na1 position. Similarly, there is a shoulder in the long bond of Na1-O3 with harmonic curves for the short Na1-O3 and both Na1-O4 bonds. The Na2-O*x* bonds can be seen in Figure 6d, with Na2-O2 showing a similar pattern to Na1-O1 for the long bond (cyan), albeit much more pronounced.

## 4. Discussion

The discussion will mainly focus on the deviations from the expected harmonic behavior. The positional shifts of bond minima can likely be attributed to the different peak values for each bond. The effect is intensified by the anharmonic contributions found in several bonds, e.g., the long O2-Na2 bond (cyan, Figure 6d). The shapes of the Nb-O bonds in Figure 5, especially in LN and NN, suggest that the MEM electron density distribution contains information about the hybridization behavior with the Nb ions ’leaking’ electron density toward the bond, thus creating an asymmetrical shape in the middle, which is not found in KN. Furthermore, with all KN Nb-O bonds being harmonic and all NN Nb-O bonds being anharmonic, LN is the only material with harmonic and anharmonic Nb-O bonds. This observation may hint at the possibility of LN to form an AFE structure, if the described features of the Nb-O bonds where sufficient conditions for AFE (see discussion below). The A-O bond anomalies in Figure 6a–d deliver further information. For all three investigated materials, one of the longer A-O bonds features an A-sided shoulder, indicating an interaction that differs from the other, more harmonic, A-O bonds. The shoulders appear to be more prevalent in LiNbO_3_ and NaNbO_3_ than in KNbO_3_ and there is a significant difference between Na1 and Na2 in NN. The strongest occurrences of A-O interaction are the local maximum in the Li-O bond as well as the large shoulder in the long Na2-O2 bond. The anharmonic character and proximity to the A-site ion fit the observations Zhang et al. made in their theoretical AFE CaTiO_3_ work [17]. Although the anharmonic interaction can also be found in KN, the effect is much weaker in comparison to the other materials, meaning the anharmonic contribution may be too weak to take effect or hindered by the size of the K ion.

Taylor expansions are performed around the A-side shoulders of the A-O bonds to find the leading terms of the Taylor expansions, gauge the strength of the anharmonicity in the measured niobates and compare the A-O interaction found in this work to the fourth order anharmonic potential term Zhang et al. used to obtain AFE CaTiO_3_. The Taylor expansions (dashed lines) plotted in Figure 7 are terminated after the 6th order term, which is required to get a decent fit for the entire bond, but locally the expansions yield good agreement terminating the expansions after the fourth order term, showing the anharmonic character of the observed A-O interaction (All numbers rounded to second decimal. The value at the peak of the Na2-O2 shoulder is off by about 0.1 e/Å^3^ due to the initial polynomial fit, which is used to obtain a continuous function and thus circumvent the issue of uneven voxel spacing along the bond line). Dividing the Taylor coefficients by the respective factorials shows that the fourth order term is the leading term in all three Taylor expansions in Equations (Equation 5), (Equation 6) and (Equation 7). Equations (Equation 6) and (Equation 7) can be found in the Appendix A. As expected, the fourth order term is most dominant over the lower order terms in the Na2-O2 bond.(5)$$\begin{array}{cc} \rho_{MEM,NN}\left(x\right)= & 0.66+0.22\left(x-2.15\right)-\frac{2.50}{2!}{\left(x-2.15\right)}^{2}\\ \; & +\frac{25.42}{3!}{\left(x-2.15\right)}^{3}+\frac{338.22}{4!}{\left(x-2.15\right)}^{4}\\ \; & \left[+\frac{1524.05}{5!}{\left(x-2.15\right)}^{5}+\frac{2707.00}{6!}{\left(x-2.15\right)}^{6}\right] \end{array}$$

Regarding the origin of antiferroelectricity, let us consider the necessity and sufficiency of criteria. Nb-O hybridization is a necessary criterion for both ferroelectricity and antiferroelectricity, but cannot be used as a sufficient marker for either structure if the Goldschmidt tolerance factor is slightly below one. Instead, the existence of a ‘strong’ anharmonic A-O interaction is implemented in this work as a sufficient criterion for antiferroelectricity in alkali niobates, which is supported by the results of a recent study by Yoneda et al., which show the Nb-O bonding in niobate perovskites being influenced by A-O hybridization differences between NN, KN, and AN [16]. The Goldschmidt tolerance factor being slightly lower than one may very well be another necessary condition for antiferroelectricity, but without an additional interaction parameter, e.g., the anharmonic A-O interaction presented in this study, the resulting structures should be FE. The importance of the Goldschmidt factor becomes more prevalent when looking at the implications of our data. Our data suggests that the KNN solid solution should result in a FE phase due to a lack of anharmonic interaction, as also suggested by the Goldschmidt factor. On the other hand, even when assuming stronger anharmonicity in Li than in Na, substituting with Li may initially stabilize the AFE phase, but the AFE structure will become less stable than the rhombohedral FE structure shared by LN and NN at low temperature. As Yoneda et al. have shown, substituting as little as 6% Li into NN leads to coexistence of the P and R phases of NN [29].

The anharmonic contribution to the NN potential found in this work could serve to explain the invar effect found in temperature dependent X-ray studies by Reznichenko et al., who reported an area of incommensurate phases at slightly elevated temperatures [30].

Comparing our results to those of other methods, Raman spectra of NN show no clear indication of the anharmonic A-O interaction apart from the evidence of phase transitions, due to the small magnitude of the described effect. The broadening of a Raman peak influenced by it would be minimal and likely very hard to distinguish from an idealized peak. In their low temperature Raman study of NN, Mishra et al. argue that missing Raman modes in their spectra may be caused by wavenumber degeneracy or small polarisability, possibly further increasing the difficulty of measuring the A-O interaction presented in this work using Raman spectroscopy [31].

Comparing our findings to other perovskite materials, Kuroiwa et al. used MEM-based EDs, which could carry information of Pb-O anharmonic interaction, to confirm the absence of Pb disorder in the AFE phase [6], but the lone pair found in Pb would likely overshadow this effect by at least one order of magnitude. On the other hand, a paraelectric perovskite, like BaZrO_3_, would have possible anharmonic interaction overshadowed by the A-site disorder commonly associated with paraelectric perovskites as explained in [32], also confirmed for the paraelectric phase of PZ in Kuroiwas ED study.

## 5. Conclusions

In summary, the MEM electron densities of LN, KN, and NN have been calculated from powder X-ray diffraction experiments. By visualizing the MEM electron density values between neighboring atoms, significant differences have been found in the strength of the anharmonic character of the A-O bonds. This anharmonic character is most pronounced in NN, leading to the conclusion that anharmonic interactions could lower the total energy to favor an antiferroelectric structure over a ferroelectric structure. This observation adds to the known criteria of B-O hybridization and the Goldschmidt tolerance factor *t*, by contributing a possible reason for irregularities known to occur just below t=1. Further investigations will need to be carried out regarding the origin of the described anharmonicity, and whether the potential to form antiferroelectric phases, as described for LN above, is much more common than the actual formation of such structures in low tolerance factor perovskites. If the anharmonic A-O interaction behavior could be predicted, the design of solid solutions that stabilize AFE phases would receive a massive boost, resulting in a step towards a better utilization of the AFE effect in commercial electronics. Since this work provides first experimental evidence for the importance of anharmonic potential contributions in antiferroelectrics, we hope to encourage theoretical researchers to further study the A-O interaction behavior in perovskites.

## Figures and Tables

**Figure 1 materials-18-04593-f001:**
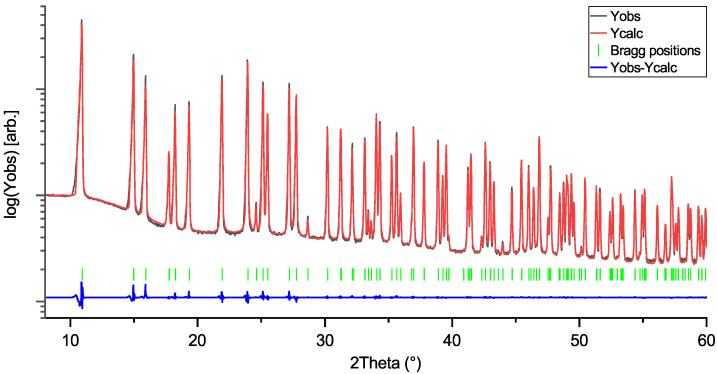
Rietveld profile of LiNbO_3_ (LN) measurement. Space group: R3ch; R = 3.20; R_f_:1.81.

**Figure 2 materials-18-04593-f002:**
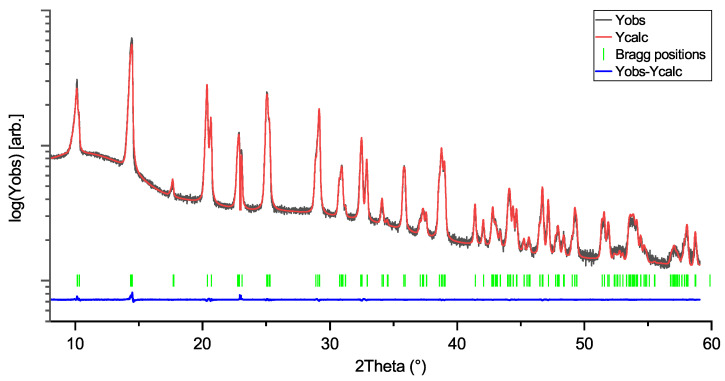
Rietveld profile of KNbO_3_ (KN) measurement. Space group: Amm2; R = 4.96; R_f_:3.69.

**Figure 3 materials-18-04593-f003:**
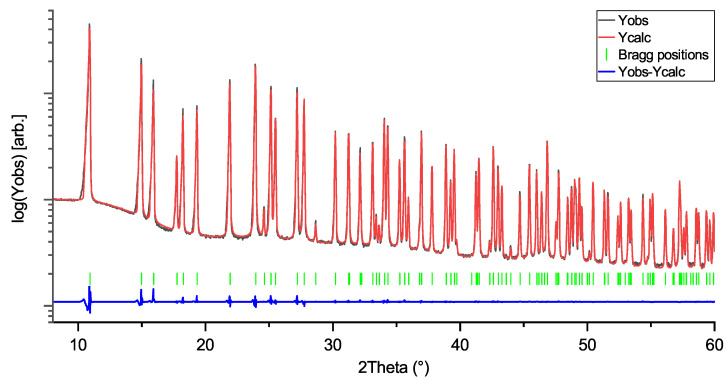
Rietveld profile of NaNbO_3_ (NN) measurement. Space group: Pbcm; R = 11.8; R_f_:6.3.

**Figure 4 materials-18-04593-f004:**
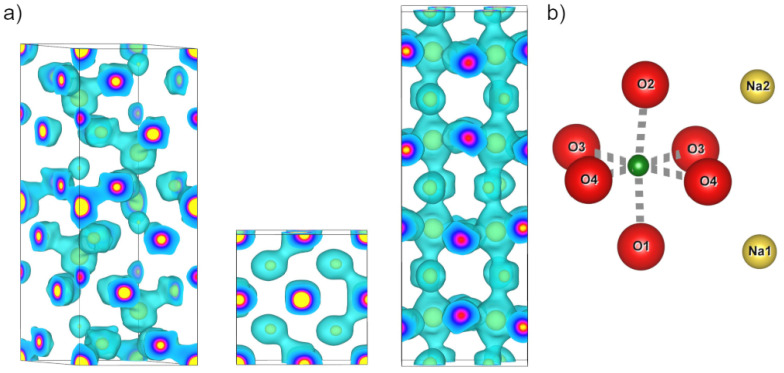
(**a**) Left to right: Maximum entropy method (MEM) electron densities (ED) in the unit cells of LiNbO_3_, KNbO_3_, and NaNbO_3_ at an isosurface level of 1 e^−^/Å^3^ (cyan). Yellow isosurface at 14 e^−^/Å^3^ for better visibility of atomic positions and depth perception. (**b**) Positional reference for NN: O*x* (red), and Na*x* (yellow) and Nb (green).

**Figure 5 materials-18-04593-f005:**
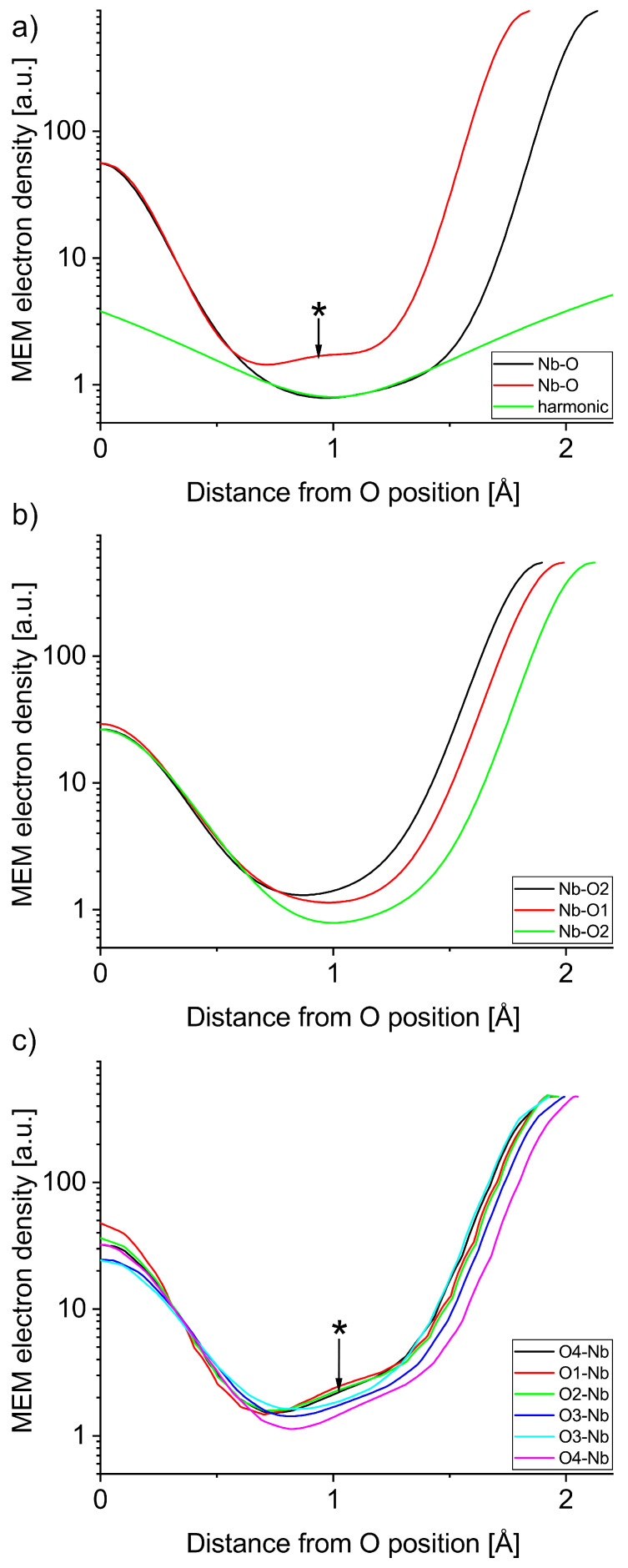
MEM electron density profiles of Nb-O bonds in (**a**) LN and exemplary harmonic curve, (**b**) KN, and (**c**) NN. Features marked with an asterisk are discussed in the text.

**Figure 6 materials-18-04593-f006:**
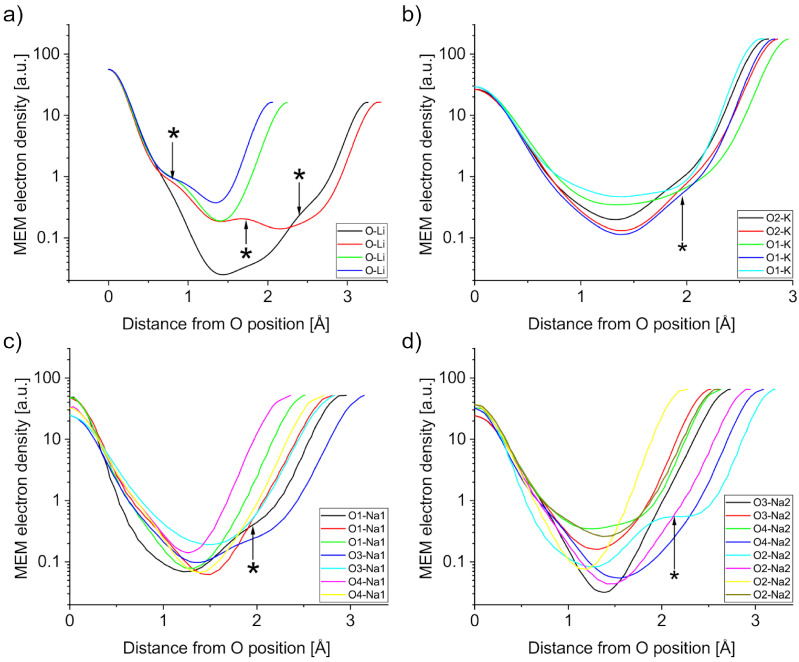
MEM electron density profiles of A-O bonds in (**a**) LN, (**b**) KN, (**c**) and (**d**) NN. Features marked with an asterisk are discussed in the text.

**Figure 7 materials-18-04593-f007:**
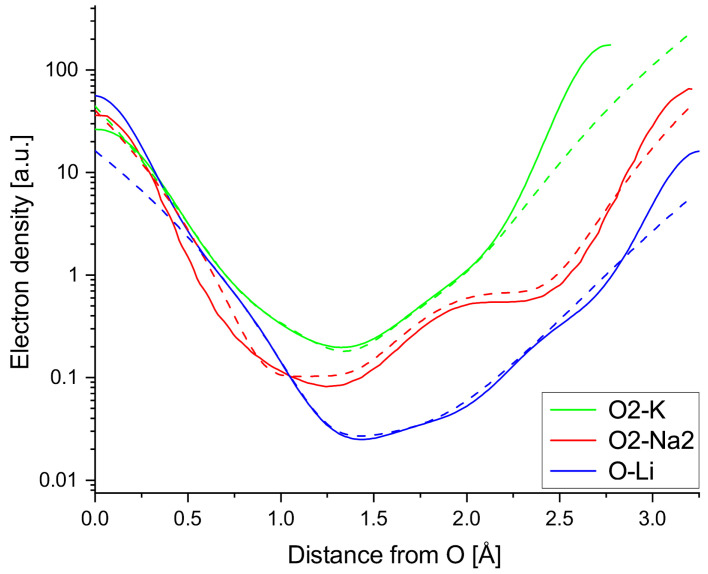
Taylor expansions (dashed lines) of A-O bonds terminated after the sixth order compared to measured ED line profiles (solid lines) of the chosen A-O bonds.

## Data Availability

The data presented in this study are available in TUdatalib at 10.48328/tudatalib-1890.2.

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
