# Peer review of "The Role of Anharmonicity in (Anti-)Ferroelectric Alkali Niobates"

_materials, 2025, doi:10.3390/ma18194593_

Round 1
Reviewer 1 Report
Comments and Suggestions for Authors
The authors investigate the origin of antiferroelectricity in NaNbO3 and its comparison with ferroelectric counterparts LiNbO3 and KNbO3 using experimental electron density (ED) mapping via the MEM based on X-ray diffraction. The authors explore the hypothesis that anharmonic A–O interactions are key to antiferroelectric behavior and present a line profile analysis of electron density between cation-anion pairs. The manuscript contributes novel experimental insights into the debated role of A-site behavior in phase instabilities and electric ordering in ABO3 perovskites. Minor revision is suggested before publishing of this manuscript to Materials.
1) How do the authors disentangle the role of A–O anharmonicity from other known structural drivers of antiferroelectricity in NaNbO3, such as octahedral tilting and NbO₆ distortions? Could the observed anharmonicity be a secondary effect resulting from broader structural modulations?
2) Have the authors considered the possible contribution of local or dynamic disorder such as off-centering of Nb or competing polar/nonpolar order parameters that may mimic apparent anharmonicity in electron density line profiles?
3) Since phonon anharmonicity is a core theme, Raman spectroscopy would be very valuable addition to this study. Temperature-dependent Raman scattering would allow direct observation of phonon mode broadening, asymmetry or splitting that supports the claim of anharmonic potential wells. Even reference to prior Raman studies would strengthen the argument. And if they study temperature dependent data, then it is to note that NaNbO3 is known to undergo complex phase transitions (orthorhombic to tetragonal to cubic) with temperature. Can the observed A–O bond profile changes be entirely decoupled from these phase transitions?
4) I suggest authors to compare their resutls with other known perovskites from lietrature. For example, how would the electron density profiles of PbZrO3 (a classical antiferroelectric) or BaZrO3 (a centrosymmetric, paraelectric perovskite) behave under similar MEM analysis?
Author Response
We thank the reviewer for helping us improve the article by providing valuable questions readers might ask upon reading the article.
1) Changed the introduction to emphasize interplay of the Pseudo-Jahn-Teller effect, changes in the potential and octahedral tilts and distortions. The anharmonicity found in the A-O bonds merely serves as the contribution to the potential that tips the scales in favor of an antiferroelectric structure, thereby causing the tilts and distortions to change accordingly.
2) Local disorder like Nb off-centering is well described by the Pseudo-Jahn-Teller effect for ferroelectric structures and should work isometrically in antiferroelectric structures, as stated by Bersuker et al. (reference added). Generally, the positional shifts and resulting polarization are caused by changes in the potential due to hybridizations, which in turn introduce bond anharmonicities. Thus, change of an order parameter is a direct expression of anharmonicity in perovskite antiferroelectrics.
3) To enhance our argument of an anharmonic force causing antiferroelectricity in niobates, we added a short literature discussion of temperature dependent x-ray and Raman data for NaNbO3 and the possibility of using Raman to confirm the interaction found in this study near the end of the discussion.
4) To address possible question as to why this effect is not generally found in all perovskite niobate data, we added a paragraph to the end of the discussion detailing the observability of our findings in the prototype systems PbZrO3 and BaZrO3.
Reviewer 2 Report
Comments and Suggestions for Authors
I think this manuscript makes an interesting and valuable contribution to the discussion of ferroelectric and antiferroelectric behavior in alkali niobates. The authors use powder X-ray diffraction with Rietveld refinement and the Maximum Entropy Method to analyze electron density distributions. I believe the main strength of the paper lies in its attempt to connect experimental electron density maps with the theoretical concept of anharmonic A–O interactions as a possible sufficient condition for antiferroelectricity. The results show that while Nb–O hybridization is present across all samples, only NaNbO3 exhibits strong anharmonic A–O contributions, which could explain its unique antiferroelectric ground state compared to the ferroelectric nature of the other niobates. In my opinion, the Taylor expansion of the MEM line profiles adds credibility to this interpretation by directly showing higher-order anharmonic terms, and the study has potential implications for designing materials with tunable ferroelectric or antiferroelectric phases.
That said, I think the novelty of the study could be clarified more explicitly. The introduction reviews prior theories and computational work, but it is not always clear what is gained experimentally here beyond confirming the role of anharmonicity. In my opinion, the manuscript would be stronger if the authors could highlight more clearly how their use of powder-based MEM goes beyond earlier MEM or ab initio studies.
The quantification of anharmonicity could also be developed further. The Taylor expansion is a useful tool, but in my opinion, the argument would be stronger if the authors compared the relative strength of anharmonic terms systematically across all three niobates. This would help substantiate the claim that NaNbO3 is the most anharmonic.
I also believe more discussion on uncertainties and robustness would strengthen the work. MEM reconstructions are sensitive to dataset quality, and while the authors acknowledge this, I think it would help to provide error estimation or a discussion of reproducibility. For example, whether multiple NaNbO3 crystals yielded the same anharmonic features or not?
In my opinion, the discussion of implications for materials design could also be expanded. The authors mention the possibility of tuning antiferroelectric phases, but the paper would benefit from more concrete examples. For instance, would partial substitution of Na with K or Li stabilize or destabilize the antiferroelectric phase? I think readers interested in applications would appreciate such elaboration.
Finally, the figures could be improved for clarity. The anomalies in Figures 5 and 6 are important for the argument, but they are not as visually clear as they could be. Zoom-ins or overlays with harmonic fits would help readers see the deviations more easily. Adding short explanatory notes or simplifying some of the terminology would improve this.
Author Response
We thank the reviewer for their suggestions towards improving the robustness, quality and accessibility of our work.
1) We added a sentence to the methods section clarifying why powder diffraction is preferable over single crystal diffraction in the case of heavily twinned single crystals. Powder MEM is not a new approach, and the main focus of this work is the confirmation of the A-O bond anharmonicity previously suggested in theoretical work.
2) To strengthen our argument that the anharmonic contribution is the most prominent in NaNbO3, we performed Taylor expansions for one anharmonic bond per material and changed figure 7 and the respective text accordingly. Excluded the long Li-O bond, because the bump in the middle does not necessarily represent A-side A-O anharmonicity.
3) A similar anharmonic bonding feature has been found in single crystal NaNbO3 data, another NaNbO3 powder sample and a second measurement of the same sample. The single crystal ED looks less trustworthy due to artifacts, especially in the Na2-O2 plane. That is why powder data was used.
4) To give readers an idea of the implications of our work, we added three sentences to the end of the discussion talking about the influence of anharmonicity and its interaction with the Goldschmidt factor with respect to solid solutions and K/Li substitution in NaNbO3.
5) Added an exemplary harmonic bond fit to LiNbO3 Nb-O profile for easier understanding of data and changed the text at the beginning of results section accordingly
Reviewer 3 Report
Comments and Suggestions for Authors
The authors measure powder X-ray diffraction (Mo-Kα1) for LiNbO₃, KNbO₃, and NaNbO₃, extract structure factors via Rietveld refinement, and compute Maximum Entropy Method (MEM) electron densities.
Major comments and questions:
- Lines ~119–121 (x-weighting 2–3): Can you show that the bond shoulder features are insensitive to x within 2–4? A small panel with overlays would help.
- Lines ~126–131 (IRF and R values): You note that omitting the IRF improves R for NN, but keep it for physical correctness. How sensitive are the MEM shoulders to using vs omitting IRF?
- Lines ~147–170 (bond profiles): How were bond lines defined (atom–atom straight line, or along the bond path from ED topology)? Did you check whether using the bond path (gradient path) changes the presence/size of shoulders?
- Lines ~194–202 (Taylor expansion / Eq. 4): Since Eq. 4 is a fit to ED, not to potential energy, can you justify interpreting the 4th-order term as “anharmonic interaction” rather than “ED curvature effect”? A DFT-validated mapping here would be ideal.
- Lines ~221–227 (design implications): Could you propose a quantitative predictor (e.g., threshold value of your asymmetry index) that flags likely AFE behavior in A-site-modified solid solutions?
- Ref. 10 (Collins, 1982) lacks the journal name.
- The introduction relies on rather outdated references, which raises doubts about the novelty and relevance of the work. It would be beneficial to add more recent references.
Author Response
We thank the reviewer for their insights and criticism, helping us to improve the clarity and overall quality of our work.
1) Added an image to the appendix showing the effect of the x parameter on the long Na2-O2 bond to strengthen trust in convergence and show that the described anharmonicity persists when giving the algorithm a stricter set of structure factors.
2) Not using the instrumental resolution file (IRF) results in a similar electron density (ED) with equally similar bond shapes, however the ED, especially around the O positions appears less well defined, leading us to believe that the Rietveld refinement is slightly overfitted. The shoulder is not significantly affected by the IRF.
3) Added the information that bond profiles are read peak to peak to the methods section. The big shoulder in the Na2-O2 bond would be a bit less prominent going along the exact gradient line, whereas the gradient lines hardly differ from the peak to peak lines for the Nb-O bonds.
4) Added a sentence to the introduction relating the Maximum Entropy Method ED to the electrostatic potential. Also added a sentence to the end of the conclusion encouraging theoretical groups to calculate and confirm our findings, which would likely also result in an answer to 5)
5) A quantitative predictor is very hard to accurately determine from our data, since the effect appears to be very small in comparison and our data more accurately describes positional information than exact magnitude, especially because we can’t quantitatively distinguish between different potential contributions. The best way to obtain a quantitative threshold would likely be in-depth DFT work varying the amplitude of the 4th-order anharmonic term Zhang et al. used, as well as the ferroelectric base material it is applied to.
6) fixed missing journals for some references
7)Added some more recent literature to the introduction to ensure readers that the problem is relevant.
Round 2
Reviewer 3 Report
Comments and Suggestions for Authors
Although the paper may be recommended for publication, I would still like to ask the authors: is it really so difficult to prepare the references according to the journal’s guidelines, where each citation should include the year, volume, and either article number or page range?
You are preparing an article for publication and at the same time ignoring the obvious rules of the journal.
Author Response
The authors are sorry about forgetting to check the new references for proper formatting. We hope to have corrected the references in question to the reviewers satisfaction.